# Novel Metallo-Supramolecular Polymers with 1-Thioxophosphole Main-Chain Units and Remarkable Photoinduced Changes in Their Resonance Raman Spectra

**DOI:** 10.3390/polym14235207

**Published:** 2022-11-30

**Authors:** Ivana Šloufová, Tereza Urválková, Muriel Hissler, Jiří Vohlídal

**Affiliations:** 1Deptartment of Physical and Macromolecular Chemistry, Faculty of Science, Charles University, 128 40 Prague 2, Czech Republic; 2CNRS, University Rennes, ISCR-UMR 6226, 35000 Rennes, France

**Keywords:** coordination polymer, intramolecular charge transfer, iron(II) metallo-supramolecular polymer, metal-to-ligand charge transfer, MLCT, phosphole, terpyridine

## Abstract

New low-bandgap unimers, with the central thiophene-(1-thioxophosphole)-thiophene (TPT) ring sequence and 2,2′:6′,2″-terpyridin-4′-yl (*tpy*) end groups connected to the central unit via conjugated linkers of different size, are prepared and assembled with Zn(II) and Fe(II) ions to metallo-supramolecular polymers (MSPs) that are studied regarding their properties. The most interesting feature of Zn-MSPs is the luminescence extended deep into the NIR region. Fe-MSPs not only show the metal-to-ligand charge transfer (MLCT) manifested by the MLCT band (an expected feature) but also an as-yet-undescribed remarkable phenomenon: specific damping of the bands of the TPT sequence in the resonance Raman spectra taken from solid Fe-MSPs using the excitation to the MLCT band (532 nm). The damping is highly reversible at the low laser power of 0.1 mW but gradually becomes irreversible as the power reaches ca. 5 mW. The revealed phenomenon is not shown by the same Fe-MSPs in solutions, nor by Fe-MSPs containing no phosphole units. A hypothesis is proposed that explains this phenomenon and its dependence on the irradiation intensity as a result of the interplay of three factors: (i) enhancement of the MLCT process by excitation radiation, (ii) the electron-acceptor character of the 1-thioxophosphole ring, and (iii) morphological changes of the lattice and their dependence on the population of new structures in the lattice.

## 1. Introduction

Soft semiconducting organic and organic/inorganic materials for applications in photovoltaics, optoelectronics, and molecular electronics are of permanently high interest [1,2,3,4,5,6]. In the last decades, the variety of these materials has grown to include metallo-supramolecular polymers (MSPs). Molecules of these materials are composed of properly designed, mostly organic low-molar-mass building blocks with chelate end-groups (referred to as unimers [7]) and metal ions in an alternating arrangement. Since MSPs are typically formed by the spontaneous self-assembly of the mentioned species through reversible metal–ligand coordination, their stability is effectively controlled by thermodynamics. As a result, the MSP chains can spontaneously partially or completely disassemble at elevated temperatures or in a solution; in this state, they exchange ions and/or unimer units with the environment and reassemble into structurally the same or a different MSP after cooling and/or drying. Therefore, these polymers are called constitutional dynamic polymers, or dynamers [8]. Structural dynamics make MSPs promising materials for the development of functional self-organized architectures, with their tunable properties, self-healing capability, and environmental adaptability. In addition, the incorporation of metal complex linkages into polymer chains has opened new approaches for tuning their magnetic, redox, electrochromic, and optical properties [3,4,5,6,9,10,11,12,13,14,15].

The dynamics and stability of MSP chains are basically tuned by varying the combinations of metal–unimer end groups. The optoelectronic and other properties are further tuned by varying the structure of the unimer’s central parts, which should enable the transfer of charge and/or energy. Of course, coordination linkers also contribute significantly to these properties. The tridentate 2,2′:6′,2″-terpyridin-4-yl group (*tpy*) is an attractive chelate end-group for conjugated MSPs, because it forms well-defined octahedral complexes with a wide range of transition metal ions through dπ–pπ* bonding, which allows electronic communication between the complexed units [2,16]. The kinetic stability of [M(*tpy*)_2_]^2+^ links (M stands for metal(II) ions) ranges from labile (e.g., M = Zn^2+^ or Cu^2+^) to almost stable (e.g., Fe^2+^ and Ru^2+^) [17]. The kinetic stability is often referred to as constitutional dynamics [8], e.g., fast for [Zn(*tpy*)_2_]^2+^ and slow for [Fe(*tpy*)_2_]^2+^ links [8,18,19]. The [M(*tpy*)_2_]^2+^ links are also of high importance for the luminescence of conjugated MSPs, which can be strong for those ones with, e.g., [Zn(*tpy*)_2_]^2+^ links but practically quenched for those with [Fe(*tpy*)_2_]^2+^ links. 

A high variety of linear-conjugated MSPs have been prepared by assembling Ru^2+^, Fe^2+^, Zn^2+^, Co^2+^, and other ions with α,ω-bis(*tpy*) (het)arylene and oligo(het)arylene unimers containing main-chain phenylene- [16,20,21,22,23,24], fluorene- [20,25,26,27], and thiophene-based [13,28,29,30,31,32] constitutional units. These units are, however, composed of σ-bonded highly aromatic rings, which do not enable an efficient delocalization of π-electrons owing to the rather high autonomy of aromatic systems. Therefore, the incorporation of sufficiently stable units with lowered aromaticity into unimers had become desirable, and the phosphole ring has been found to be a good candidate for this purpose [33,34,35,36,37,38]. The phosphorus atom in a P-substituted ring acquires pyramidal geometry in which its lone electron pair hardly interacts with the endocyclic diene π-electrons [35,37,38]. Therefore, the delocalization of electrons in the phosphole ring stems from the so-called σ–π hyperconjugation involving the exocyclic P-R σ-bond and the endocyclic diene π-system [35,37]. As a result, phosphole is less aromatic compared to pyrrole and thiophene [37,39,40]. Mixed phosphole–thiophene derivatives are widely investigated phosphole-based π-conjugated systems with low band-gap energies and a high potential of structural variations, by the choice of the P-atom substituent R and the oxidation of the ring P-atom to P = S or P = O [36,41,42,43,44]. We recently reported a new α,ω-bis(*tpy*) unimer (**TPT**) containing a central unit composed of a 1-thioxophosphole- -2,5-diyl central ring surrounded by two thiophene-2,5-diyl rings and showed that the band-gap energy of this new unimer is significantly lower compared to the energy of α,ω-bis(*tpy*)terthiophene [19]. This proved that replacing the central thiophene-2,5-diyl unit with a low-aromaticity phosphole unit significantly increased the delocalization of the electrons in the unimer as well as in its MSPs, compared to the corresponding bis(*tpy*)terthiophene unimers and their MSPs (1-thioxophosphole unit caused a red shift relative to bis(*tpy*)terthiophene unimers of almost 100 nm). In the present paper, we report the preparation and characterization of a series of new conjugated low-bandgap bis(*tpy*) unimers with a **TPT**-type central unit and different linkers connecting *tpy* end-groups to this unit (see Figure 1), assembly of these unimers with Fe^2+^ and Zn^2+^ ions into corresponding metallo-supramolecular polymers (Fe-MSP and Zn-MSP), and characterization of these MSPs. Finally, we report an as-yet-unknown and remarkable phenomenon that exhibited all new Fe-MSPs derived from unimers with a 1-thioxophosphole unit: reversible photoinduced selective attenuation of the Raman bands of the central **TPT**-type central unit in the resonance Raman spectra excited to the metal-to-ligand charge transfer band (MLCT band).

## 2. Materials and Methods

**Materials.** Isopropylmagnesium chloride (2.0 M solution in diethylether, concentration checked by titration), 2-bromo-4-hexylthiophene, *N*-iodosuccinimide (NIS), 1,7-octadiyne, copper iodide, diisopropylamine, bis(triphenylphosphine)palladium(II) dichloride (Pd(PPh_3_)_2_Cl_2_, tetrakis(triphenylphosphine)palladium(0) (Pd(PPh_3_)_4_)_,_ titanium(IV) isopropoxide, sulphur, zinc(II) perchlorate hexahydrate, iron(II) perchlorate hydrate (all Aldrich), potassium carbonate, magnesium sulphate (VWR), 4′-(4-bromophenyl)-2,2′:6′,2″-terpyridine, and 4′-bromo-2,2′:6′,2″-terpyridine (TCI) were used as received. *P*,*P*-dichlorophenylphosphine (Aldrich) was distilled trap-to-trap before use. Hexane (Lachner) and acetonitrile (ACN) (Aldrich) were stored over molecular sieve; tetrahydrofurane (Aldrich) was distilled from LiAlH_4_ or sodium/benzophenone before use or obtained from the MBraun drying solvents system (SPS-800); toluene (Lachner) was distilled from sodium/benzophenone before use; methanol was bubbled with Argon before use; diethylether, pentane, dichloromethane, chloroform (Lachner), and acetonitrile (ACN) were used as obtained.

**Instrumentation.**^1^H and ^13^C spectra were recorded on a Varian ^UNITY^INOVA 400 or Varian SYSTEM 300 instrument in CD_2_Cl_2_ or CDCl_3_ and referenced to the solvent signal: 7.25 ppm (CDCl_3_) or 5.32 ppm (CD_2_Cl_2_) for ^1^H spectra and 77.0 ppm (CDCl_3_) or 53.84 ppm (CD_2_Cl_2_) for ^13^C spectra. ^13^C NMR spectra of unimers were recorded on Bruker AVANCE III (600 MHz) (Bruker Corporation, Billerica, MA, USA). Coupling constants, *J* (in Hz), were obtained by the first-order analysis. The UV/vis spectra were recorded on a Shimadzu UV-2401PC instrument (Shimadzu Corporation, Kyoto, Japan); photoluminescence spectra were recorded on a Fluorolog 3-22 Jobin Yvon Spex instrument (Horiba, Kyoto, Japan), using four-window quartz cuvette (1 cm) for solutions and quartz glass support for films. The photoluminescence absolute quantum yields were determined using an integration sphere Quanta-φ F-3029. Raman spectra were recorded on a DXR Raman spectrometer (Thermo Fischer Scientific, Waltham, MA, USA) interfaced to an Olympus microscope (objective 50×), using full-range gratings (3300–40 cm^−1^, 400 lines/mm) and thin films of Fe-MSPs deposited on glass slides.

### 2.1. Synthesis

Synthetic pathways to unimers are presented in Figure 2. In summary, the central unit of unimers, 2,5-bis(4-hexylthiophen-2-yl)(1-phenyl-1-thioxo)phosphole, was prepared by using the Sato protocol [45] and procedure described earlier [19] and then iodinated using *N*-iodosuccinimide. The linkers’ precursors caped by *tpy* group, 4′-(ethynylaryl)terpyridines, were prepared using the procedures described in [46,47,48] and connected to the central unit by the Sonogashira coupling.

### 2.2. Precursors of Linkers

**4**′**-Ethynyl-2,2:6,2-terpyridine** was prepared from commercial 4′-bromo-2,2:6,2-terpyridine by the procedure described in [46,47]. Other precursors, **4**′**-(ethynylaryl)-2,2:6,2-terpyridines**, were prepared by Sonogashira coupling of the corresponding bromoderivatives with (triisopropylsilyl)acetylene followed by the deprotection of ethynyl group [47,48]. Briefly: 4′-(bromoaryl)terpyridine (Equation (1)), copper iodide (2 mol %), and Pd(PPh_3_)_2_Cl_2_ (2 mol %) were dissolved in THF/triethylamine (1:1 by vol.) mixed solvent, and (trimethylsilyl)acetylene (Equation (2)) was added. The reaction mixture was heated to 60 °C, and the course of the reaction was monitored by the gas chromatography. After the starting materials were consumed, the volatiles were evaporated, the solid crude product was dissolved in diethyl ether, and the obtained solution was filtered through basic alumina. The purified intermediate was then dissolved in a THF/methanol mixed solvent and mixed with potassium carbonate to remove the TMS groups protecting the ethynyl groups.

4′-(4-Ethynylphenyl)-2,2:6,2-terpyridine was prepared from commercial 4′-(4-bromophenyl)-2,2:6,2-terpyridine. 4′-(5-Ethynylthiophen-2-yl)-2,2:6,2-terpyridine and 4′-(5-ethynyl-3-hexylthiophen-2-yl)-2,2:6,2-terpyridine were synthesized from corresponding bromo-derivatives prepared by the procedure described in [18,48]. The spectroscopic characteristics of the linker precursors, except for the last one (see next paragraph), are available in the above indicated references.

**4**′**-(5-ethynyl-3-hexylthiophen-2-yl)-2,2**′**:6**′**,2″-terpyridine** was obtained as a yellowish solid in the isolated yield of 55% by the above-described procedure.

^1^H NMR (300 MHz, CD_2_Cl_2_, δ/ppm): 8.69–8.73 (m, 2H, A6), 8.67 (d, *J* = 7.93 Hz, 2H, A3), 8.57 (s, 2H, B3), 7.90 (td, *J* = 7.00, 1.70 Hz, 2H, A4), 7.37 (ddd, *J* = 7.45, 4.77, 1.29 Hz, 2H, A5), 7.25 (s, 1H, thiophene), 3.49 (s, 1H, ethynyl), 2.78 (t, *J* = 8.10 Hz, 2H, Hex1), 1.62–1.75 (m, 2H, Hex2), 1.48–1.50 (m, 2H, Hex3), 1.24–1.32 (m, 4H, Hex4 + Hex5), 0.82–0.87 (m, 3H, Hex6).

HRMS found m/z 423.5725 [M + H^+^], and C_27_H_26_N_3_S requires 423.58.

FT-IR (cm^−1^): 3217 (m), 3172 (s), 3072 (m), 3062 (m), 3049 (m), 3013 (m), 2954 (s), 2931 (s), 2868 (m), 2854 (s), 1602 (s), 1585 (s), 1568 (s), 1557 (s), 1539 (s), 1466 (s), 1435 (m), 1394 (s), 1378 (m), 1267 (m), 1124 (m), 1096 (w), 1089 (m), 1073 (m), 1026 (m), 989 (m), 964 (w), 894 (m), 853 (m), 793 (s), 777 (s), 751 (s), 742 (s), 676 (s), 658 (m), 617 (s), 587 (m), 549.61 (w), 403 (w).

### 2.3. Precursor of the Central Unit

**2,5-bis(4-hexylthiophen-2-yl)(1-phenyl-1-thioxo)phosphole (1)** was prepared by using the Sato protocol [45] and a modified procedure described earlier for the compound without hexyl groups [19]. A solution of 2,2′-(octa-1,7-diyne-1,8-diyl)bis[4-hexylthiophene] (0.55 g, 1.25 mmol) in diethyl ether was tempered to −50 °C, and 1.1 equivalent of Ti(OiPr)_4_ (0.4 mL, 1.375 mmol) and 2.1 equivalent of ^i^PrMgCl (1.4 mL; 2.62 mmol) were added. The mixture kept at −50 °C for 2 h and changed its color from yellow to dark orange. Then, PhPCl_2_ (0.18 mL, 1.3 mmol) was added to in situ formed titanocene; the mixture was kept at −50 °C for next 20 min, then at 0 °C for 1 h, and finally at room temperature monitored by ^31^P NMR. After PhPCl_2_ was consumed and intermediates were absent, all volatiles were evaporated, and orange fluorescent solid was diluted in DCM and filtrated over basic alumina under inert atmosphere. Then, purity of product was checked by NMR, and phosphole was immediately oxidized by addition of sulfur. Oxidation was followed also by ^31^P NMR, where incorporation of sulfur causes the up-field shift from 12 to ca 52 ppm. Resulting product was purified by column chromatography using silica gel and heptane/DCM (4/1 by vol) as a mobile phase and isolated as a yellow-orange solid with a yield of 73% (0.53 g, 0.92 mmol).

^1^H NMR (400 MHz, CD_2_Cl_2_) δ 7.88 (ddd, *J* = 14.0, 8.3, 1.5 Hz, 2H, σ_Ph_), 7.48 (ddd, *J* = 7.0, 5.3, 1.8 Hz, 1H, p_ph_), 7.42 (ddd, *J* = 8.5, 6.5, 3.0 Hz, 2H, m_ph_), 7.21 (s, 2H, th5), 6.95 (t, *J* = 1.3 Hz, 2H, th3), 2.97–2.84 (m, 4H, C-*CH_2_*-CH_2_), 2.51 (t, *J* = 7.6 Hz, 4H, hex1), 1.87 (dq, *J* = 5.1, 2.6 Hz, 4H, C-CH_2_
*CH_2_*), 1.50 (t, *J* = 7.4 Hz, 4H, hex2), 1.26 (qd, *J* = 6.1, 5.5, 2.3 Hz, 12H, hex3-5), 0.92–0.83 (m, 6H, hex6).

^13^C NMR (101 MHz, CD_2_Cl_2_) δ 145.9 (d, *J* = 21.3 Hz), 144.1, 135 (d, *J* = 17.1 Hz), 132.5 (d, *J* = 3.0 Hz), 131.2 (d, *J* = 11.6 Hz), 130.2, 129.5 (d, *J* = 5.6 Hz), 129.3 (d, *J* = 12.7 Hz), 128.4, 122.4, 32.2, 31.2, 30.8 (d, *J* = 15.5 Hz), 29.7 (d, *J* = 13.3 Hz), 29.4, 23.1 (d, *J* = 16.6 Hz), 14.4.

^31^P NMR (162 MHz, CD_2_Cl_2_, δ/ppm): 51.77.

FT-IR (cm^−1^): 3073 (w), 3062 (w), 2963 (m), 2870 (w), 1455 (w), 1435 (w), 1422 (w), 1416 (m), 1308 (w), 1292 (w), 1261 (s), 1228 (m), 1185 (m), 1094 (s), 1019 (s), 936 (w), 921 (w), 908 (m), 863.47 (m), 799 (s), 749 (m), 740 (m), 696 (m), 667 (m), 641 (w), 626 (m), 606 (w), 579 (w), 565 (m), 540(m), 517 (m), 509 (m), 478 (m), 460 (w), 437 (w), 402 (m).

HRMS found m/z [M + H^+^] 579.234, and C_34_H_44_PS_3_ requires 579.896.

**2,5-bis(4-hexyl-5-iodothiophen-2-yl)(1-phenyl-1-thioxo)phosphole (2)**: **1** was iodinated using *N*-iodosuccinimide. A solution of **1** (0.345 g, 0.6 mmol) in DCM (20 mL) was cooled to −78 °C, and a solution of N -iodosuccinimide (0.284 g, 1.25 mmol) in ACN (1 mL) was added dropwise. The reaction mixture was stirred at −78 °C for 15 min and then allowed to warm to room temperature, at which point it was kept in the dark for 18 h. Then, volatiles were removed, and the solid red crude product was dissolved in DCM, washed with saturated Na_2_S_2_0_3_ solution and then three times with water, and finally purified by column chromatography (silica gel, heptane/DCM (4/1 *v/v*). **2** was isolated as a red powder (yield 0.339 g, 0.408 mmol, 68%). 

^1^H NMR (400 MHz, CD_2_Cl_2_, δ/ppm): 7.84 (ddd, *J* = 14.1, 8.3, 1.4 Hz, 2H, σ_ph_), 7.50 (td, *J* = 7.1, 1.8 Hz, 1H, p_ph_), 7.42 (td, *J* = 7.5, 3.1 Hz, 2H, m_ph_), 7.00 (s, 2H, th), 2.84 (tt, *J* = 5.9, 2.8 Hz, 4H, C-*CH_2_*-CH_2_), 2.44 (td, *J* = 7.4, 3.1 Hz, 4H, Hex1), 1.87 (dt, *J* = 6.9, 3.4 Hz, 4H, C-CH_2_-*CH_2_*), 1.43 (p, *J* = 7.4 Hz, 4H, Hex2), 1.33–1.17 (m, 12H, Hex3 + Hex4 + Hex5), 0.94–0.81 (m, 6H, Hex6).

^31^P NMR (162 MHz, CD_2_Cl_2_, δ/ppm): 51.36.

^13^C NMR (101 MHz, CD_2_Cl_2_, δ/ppm): 148.1, 146.5 (d, *J* = 20.9 Hz), 139.9 (d, *J* = 17.2 Hz), 132.8 (d, *J* = 3.1 Hz), 131.1 (d, *J* = 11.9 Hz), 129.7, 129.5 (d, *J* = 12.5 Hz), 129 (d, *J* = 1.7 Hz), 128.8 (d, *J* = 6.0 Hz), 128.2, 77.9, 32.3 (d, *J* = 35.8 Hz), 29.6 (d, *J* = 13.0 Hz), 29.2, 23 (d, *J* = 29.7 Hz), 14.4.

FT-IR (cm^−1^): 2948 (s), 2925 (s), 2866 (m), 2853 (s), 1464 (w), 1452 (w), 1439 (m), 1412 (m), 1298 (m), 1101 (s), 1016 (w), 846 (w), 836 (w), 825 (w), 749 (m), 719 (s), 692 (w), 666 (s), 603 (w), 522 (m).

HRMS found m/z [M + Na^+^] 853.009, and C_34_H_41_I_2_NaPS_3_ requires 853.657.

### 2.4. Coupling of Precursors to Unimers

**General procedure.** The chosen linker-*tpy* derivative (Equation (2)) and the diiodo-derivative **2** (Equation (1)) were dissolved in THF, catalytic amounts (2%) of Pd(PPh_3_)_4_ and diisopropylamine were added, and the reaction mixture was kept at 80 °C for 48 h (conversion was monitored by TLC). Then, volatiles were evaporated, and the obtained red solid was dissolved in DCM and washed with brine and water. The crude product was purified by washing with pentane and diethylether to remove the isomeric phospholene derivative (this byproduct is formed by base-catalyzed isomerization of phosphole ring in which one phosphole C=C bond is shifted to the exo position, changing cyclohexane into cyclohexene ring [38]), and by column chromatography (alumina and DCM or hexane/THF or toluene/acetone mobile phase). Purification was the most difficult job. Products were isolated as a dark red–violet powders in yields around 10% due to high losses during purification. For ^1^H NMR and ^13^C NMR spectra of unimers, see the Appendix A).

### 2.5. Unimer **E**

This unimer was successfully purified by liquid chromatography using DCM and basic alumina. Product was obtained as dark red solid with a yield of 11% (23 mg, 0.02 mmol).

^1^H NMR (400 MHz, CD_2_Cl_2_, δ/ppm): 8.78–8.71 (m, 4H, A6), 8.67 (d, *J* = 8.0 Hz, 4H, A3), 8.56 (s, 4H, B3), 8.01–7.89 (m, 6H, σ_ph_ + A4), 7.64–7.48 (m, 3H, m_ph_ + p_ph_), 7.41 (ddd, *J* = 7.6, 4.8, 1.3 Hz, 4H, A5), 7.32 (s, 2H, thiophene), 3.02 (s, 4H, C-*CH*_2_-CH_2_), 2.77 (q, *J* = 6.6, 5.8 Hz, 4H, Hex1), 1.98 (s, 4H, C-CH_2_-*CH_2_*), 1.61 (d, *J* = 15.9 Hz, 10H, Hex2 + water), 1.33 (dd, *J* = 9.2, 4.2 Hz, 12H, Hex 3-5), 0.98–0.82 (m, 6H, Hex6).

^31^P NMR (121 MHz, CD_2_Cl_2_, δ/ppm): 51.28.

^13^C NMR (101 MHz CD_2_Cl_2_, δ/ppm): 156.2, 156.1, 150.2, 149.8, 147.6 (d, *J* = 19.9 Hz), 137.4, 136.8 (d, J = 17.5 Hz), 133.5 (d, *J* = 3 Hz), 131.2 (d, *J* = 11.9 Hz), 130.9, 129.6 (d, J = 3.7 Hz), 129.6 (d, *J* = 12 Hz), 129.2 (d, *J* = 5 Hz), 128.8, 124.6, 122.5, 121.6, 119.6, 96.9, 87.2, 32.2, 31.2, 30.6, 30, 29.3, 23.2, 22.8, 14.5.

FT-IR (cm^−1^): 3058 (w), 3011 (w), 2950 (w), 2926 (m), 2857 (w), 2193 (m), 1598 (m), 1581.82 (s), 1566 (s), 1541 (w), 1466 (m), 1432 (w), 1391 (m), 1265 (w), 1116 (w), 1103 (w), 1093 (w), 1069 (w), 996 (w), 986 (w), 892 (w), 793 (s), 744 (m), 720 (w), 690 (w), 661 (m), 617 (w), 523 (w).

HRMS found m/z 1089.393 [M + H^+^], and C_68_H_62_N_6_PS_3_ requires 1090.428.

### 2.6. Unimer **EPh**

This unimer was successfully purified by liquid chromatography using basic alumina and toluene/acetone (gradient 0–2% of acetone). Product was isolated as a dark red solid with a yield of 6.5% (12 mg, 0.01 mmol).

^1^H NMR (300 MHz, CD_2_Cl_2_, δ/ppm): 8.80 (s, 4H, B3), 8.77–8.68 (m, 8H, A6+A3), 7.99–7.88 (m, 10H, A4+σ_ph_ + ph), 7.66 (d, *J* = 8.4 Hz, 4H, ph), 7.57–7.45 (m, 3H, m_ph_ + p_ph_), 7.40 (ddd, *J* = 7.5, 4.8, 1.2 Hz, 4H, A5), 7.27 (s, 2H, thio), 2.98 (s, 4H, C-*CH_2_*-CH_2_), 2.70 (t, *J* = 7.5 Hz, 4H, Hex1), 1.94 (s, 4H, C-CH_2_-*CH_2_*), 1.59 (m, 4H, Hex2), 1.31 (d, *J* = 3.9 Hz, 12H, Hex 3-5), 0.95–0.87 (m, 6H, Hex6).

^31^P NMR (121 MHz, CD_2_Cl_2_, δ/ppm): 51.23.

^13^C NMR (151 MHz, CD_2_Cl_2_, δ/ppm): 156.8 (s), 156.6 (s), 149.8 (s), 149.8 (s), 149.1 (s), 147.2 (d, *J* = 21 Hz), 139 (s), 137.5 (s), 135.8 (d, *J* = 17.7 Hz), 132.8 (s), 132.3 (s), 131.2 (d, *J* = 12.2 Hz), 129.6 (d, *J* = 6.6 Hz), 129.6 (d, *J* = 12.2 Hz), 129.4 (s), 128.8 (s), 127.9 (s), 124.6 (s), 124.5 (s), 121.7 (s), 120.4 (s), 119. (s), 98.1 (s), 84.5 (s), 32.2 (s), 30.6 (s), 29.9 (s), 29.9 (s), 29.3 (s), 23.29 (s), 22.9 (s), 14.5 (d, *J* = 5 Hz).

FT-IR (cm^−1^): 3051 (w), 2949 (m), 2929 (m), 2856 (w), 2191 (w), 1979 (w), 1970 (w), 1958 (w), 1940 (w), 1603 (m), 1584 (s), 1566 (m), 1523 (w), 1508 (w), 1466 (m), 1441 (m), 1412 (w), 1388 (m), 1097 (w), 1038 (w), 989 (w), 832 (m), 791 (s), 745 (m), 737 (w), 718 (m), 692 (w), 668 (w), 661 (w), 621 (w), 52 1(w).

HRMS found m/z 1241.4556 [M + H^+^], and C_80_H_70_N_6_PS_3_ requires 1242.6128.

### 2.7. Unimer **ET**

This unimer was successfully purified by using chromatography on basic alumina, first with the hexane/THF solvent system of increasing gradient polarity from 4/1 to 1/1 by volume and then second with DCM/1% methanol mobile phase. Product was obtained as a red–violet solid with a yield of 3% (4 mg, 0.004 mol).

^1^H NMR (600 MHz, CD_2_Cl_2_, δ/ppm): 8.73 (d, *J* = 4.4 Hz, 4H, A6), 8.69 (s, 4H, B3), 8.65 (d, *J* = 8.1 Hz, 4H, A3), 7.87–7.93 (m, 6H, A4+ σ_ph_), 7.71 (d, *J* = 3.9 Hz, 2H, C3), 7.50–7.54 (m, 2H, m_ph_), 7.44–7.48 (m, 1H, p_ph_), 7.40 (ddd, *J* = 7.5, 4.7, 1.2 Hz, 4H, A5), 7.33 (d, *J* = 3.9 Hz, 2H, C4), 7.27 (s, 2H, D3), 2.97 (br. s, 4H, C-*CH_2_*-CH_2_), 2.69 (t, *J* = 7.5 Hz, 4H, Hex1), 1.90–1.95 (m, 4H, C-CH_2_-C*H_2_*), 1.25–1.32 (m, 16H, Hex2-5), 0.87–0.92 (m, 6H, Hex6).

^31^P NMR (121 MHz, CD_2_Cl_2_, δ/ppm): 51.22 (s, 1P).

^13^C NMR (151 MHz, CD_2_Cl_2_, δ/ppm): 156.8, 156.2, 149.7, 149.5, 147.4 (d, *J* = 19.9 Hz), 144, 143.1, 137.6, 136.3 (d, *J* = 16.6 Hz), 133.7, 132.9 (d, *J* = 3.0 Hz), 131.2 (d, *J* = 11.1 Hz), 129.7 (d, *J* = 5.5 Hz), 129.6 (d, *J* = 13.10 Hz), 129.4, 128.9, 126.6, 125.1, 124.7, 121.7, 119.8, 117.4, 91.3, 88.5, 32.2, 30.5, 30.3, 30.1, 29.3, 23.1, 22.9, 14.5. 

FT-IR (cm^−1^): 3054 (w), 2964 (m), 2932 (m), 2909 (m), 2853 (w), 2180 (w), 1600 (m), 1584 (m), 1262 (s), 1096 (s), 1020 (s), 865 (m), 799 (s), 743 (m), 731 (w), 717.87 (m), 683 (w), 666 (m), 659 (w), 621 (w), 514 (w).

HRMS found m/z 1253.3685 [M + H^+^] C_76_H_66_N_6_PS_5_ requires 1254.6683.

### 2.8. Unimer **ET6**

This unimer was successfully purified by using the chromatography on basic alumina first with the hexane/THF (3/2 by volume) mixed mobile phase and obtained as a dark violet solid with a yield of 10% (10 mg, 0.007 mmol).

2,5-bis{4-hexyl-5-[4′-(5-ethynyl-3-hexylthiophen-2-yl)tpy]- thiophen-2-yl} 1-phenyl-1-thioxophosphole (ET6).

Column chromatography on basic alumina, using hexane/THF (3/2 by volume) mixed mobile phase. Fractions containing product were collected and evaporated, washed with diethyl ether/pentane (1/1, *v/v*) and gained as dark violet solid (10 mg, 10%).

^1^H NMR (600 MHz, CD_2_Cl_2_, δ/ppm): 8.71 (d, *J* = 4.04 Hz, 4H, A6), 8.67 (d, *J* = 8.07 Hz, 4H, A3) 8.58 (s, 4H, B3), 7.86–7.93 (m, 6H, A4 + σ_ph_), 7.50–7.55 (m, 1H, p_ph_), 7.44–7.49 (m, 2H, m_ph)_, 7.38 (ddd, *J* = 6.7, 5.4, 1.1 Hz, 4H, A5), 7.24 (s, 2H, C4), 7.21 (s, 2H, D3), 2.95 (br. s., 4H, C-*CH_2_*-CH_2_), 2.81(t, *J =* 7.5 Hz, 4H, C-Hex1), 2.66 (t, *J* = 7.4 Hz, 4H, D-Hex1), 1.92 (br. s., 4H, C-CH_2_-C*H_2_*), 1.65–1.78 (m, 4H, C-Hex2), 1.35–1.44 (m, 4H, D-Hex2), 1.24–1.33 (m, 20H, C-Hex3-5, D-Hex3-5), 0.80–0.93 (m, 12H, C-Hex6, D-Hex6).

^31^P NMR (121 MHz, CD_2_Cl_2_, δ/ppm): 51.20 (s, 1 P).

^13^C NMR (151 MHz, CD_2_Cl_2_, δ/ppm): 156.5 (s), 156.3 (s), 149.8 (s), 149.1 (s), 147.2 (d, *J* = 19.9 Hz), 143.9 (s), 142.2 (s), 138.2 (s), 137.4 (s), 136 (d, *J* = 17.70 Hz), 135.5 (s), 132.8 (d, *J* = 3.3 Hz), 131.1 (d, *J* = 11.1 Hz), 129.6 (s), 129.6 (s), 129.5 (d, *J* = 5.50 Hz), 129.3–129.3 (m), 129.1 (s), 128.8 (s), 124.6 (s), 123 (s), 121.6 (s), 120.9 (s), 120(s), 91.4 (s), 87.6 (s), 32.2 (s), 31.4 (s), 30.5 (s), 29.9 (s), 29.9 (d, *J* = 12.2 Hz), 29.6 (s), 29.6 (s), 29.3 (s), 23.2 (d, *J* = 2.21 Hz), 22.9 (s), 14.5 (s), 14.4 (s).

FT-IR (cm^−1^): 3176 (w), 3059 (w), 3014 (w), 2953 (m), 2930 (m), 2854 (w), 2181 (w), 1600 (w), 1583 (s), 1568 (m), 1558 (m), 1549 (w), 1532 (w), 1465 (m), 1445 (w), 1435 (w), 1395 (m), 1373 (w), 1266 (w), 1125 (w), 1097 (w), 1073 (w), 1022 (w), 989 (w), 893 (w), 888 (w), 844 (w), 839 (w), 793 (m), 743 (m), 718 (w), 687 (w), 667 (w), 659 (w), 622 (w), 616 (w).

HRMS found m/z 1421.5563 [M + H^+^], and C_88_H_90_N_6_PS_5_ requires 1421.9872.

**Assembly of Zn-MSPs and Fe-MSPs and its monitoring in solution.** For monitoring the progress in the assembly, we used the procedure described in [18,28,29]. Briefly, a measured volume of a solution of Zn^2+^ or Fe^2+^ perchlorate (2 × 10^−3^ M) in chloroform/acetonitrile mixed solvent (1/1, *v/v*) was added into a unimer (**U**) solution (2 × 10^−5^ M) in the same solvent by using a Hamilton syringe. The metal to unimer (M^2+^/**U**) mole ratio *r* (*r* = M^2+^/**U**) varied from 0 to 3. The UV/vis absorption and the photoluminescence emission spectra of the solutions were measured at room temperature one day after preparation. Films of solid MSPs for spectral measurements were prepared by mixing equimolar amounts of chosen unimer and metal salt in solution (*r* = 1), followed by the drop casting of the solution on a quartz substrate and drying the wet film. Polymers are further denoted by symbols composed of the metal symbol and the unimer abbreviation; for example, **ZnET** stands for the MSP formed from Zn^2+^ ions and unimer **ET**, and **FeEPh** stands for the MSP prepared from ions Fe^2+^ and unimer **EPh**.

## 3. Results

### 3.1. Characteristics of Phosphole Unimers and Their Polymers

The UV/vis spectral changes accompanying assembly of polymers **ZnET** and **FeET** are shown in Figure 1, and similar figures for the assembly of the other MSPs are available in Appendix A. For clarity, each spectral set is divided into two parts: the set for the ions (M) to unimer (U) mole ratios *r* = M/U from 0 to ca 0.5 (left) and the set for *r* > 0.5 (right). The main spectral changes are the following: (*i*) development of a new band at ca 330 nm, (*ii*) red shift of the HOMO–LUMO band of the unimer, (*iii*) slight narrowing but practically no change in the intensity of the band at 285 nm, and (*iv*) development of a new band at around 600 nm in the systems with Fe^2+^ ions. The stable band at 285 nm is mainly contributed to by transitions in the pyridine rings [30], and it can advantageously be used to normalize the spectra of solid MSPs (vide infra). The new bimodal UV/vis band at around 350 nm is associated with transitions within coordinated *tpy* groups that have pyridine rings in the *syn* conformation (unlike their conformation *anti* in free *tpy* groups). Essential bimodality of this band is of vibrational origin [46,49,50]. The red shift of the HOMO–LUMO band proves an increase in the delocalization of electrons upon binding unimer molecules into MSP chains. The band at around 600 nm is associated with electronic transitions relating to the metal-to-ligand charge transfer (MLCT) that is typical of [Fe(*tpy*)_2_]^2+^-type species [2,3,4,5,6,7,16,21,22,49,50,51].

As can be seen, the UV/vis spectral sets for the M^2+^/**U** systems of the mole ratios *r* from 0 to 0.5 (Figure 1, left column) showed a few isosbestic points (IP), whereas the sets for higher ratios *r* (Figure 1, right columns) did not. The presence of IPs generally indicates that defined initial species are transformed into other defined ones. Therefore, the occurrence of IPs in the spectral sets for *r* ratios up to 0.5 indicates that the new species are “butterfly” dimers **U**-M^2+^-**U**. The IP at ca 310 nm has been observed in almost all spectral sets for assembling of terpyridine and its 4′-derivatives with Zn^2+^ and Fe^2+^ ions [18,29,46,51,52]. The absence of IPs in the spectral sets for ratios *r* > 0.5 is consistent with the consecutive stepwise assembly of the preformed **U**-M^2+^-**U** species and M^2+^ ions to form [-M^2+^-**U**-]*_n_* chains of various lengths, i.e., more than one defined species. This difference between the two assembly stages proves that the *tpy* end-groups of semi-bounded unimeric units are less reactive than the *tpy* end-groups of free unimer molecules, which indicates that there is an efficient electron density transfer along the MSP chains derived from conjugated unimers.

The effects of linker structure on the UV/vis spectral characteristics of the unimer and MSPs are shown in Figure 2. As can be seen expected, the *λ*_UV_ of the unimer noticeably red-shifted, and its molar absorption coefficient substantially increased with the increasing size of the π-conjugated linker. The same *λ*_UV_ values of **ET** and **ET6** indicate practically the same band-gap energy of these unimers, despite the steric hindrances caused by the hexyl groups in **ET6.** However, these hindrances are clearly manifested by the reduced value of *ε*_UV_ for **ET6**. The *ε*_UV_ values of the unimers showed a similar trend with the increasing linker size, with *λ*_UV_ up to the dominant value for **ET**, followed by a drop of about ca. 30% to the already-justified value for **ET6**. Interestingly, the agreement of the *ε*_UV_ of **EPh** and **ET6** indicates that the steric hindrance of the hexyl side group in **ET6** quantitatively cancels the positive effect of replacing the benzene ring with a thiophene ring (the difference between **ET** and **EPh** is just canceled).

In contrast to the unimers, Zn-MSPs showed an alternating (red-shift/blue-shift) dependence of both *λ*_UV_ and *ε*_UV_, with an increasing linker size. Surprisingly, the highest *λ*_UV_ value showed **ZnE** (not **ZnET**), which, however, showed a much lower value of *ε*_UV_ compared to **ZnET**. Unlike Zn-MSPs, Fe-MSPs showed a dependence on the linker size that was similar to that of free unimers: a continuous growth of *λ*_UV_ and a growth of *ε*_UV_ up to **FeET**, followed by a drop of about ca. 25% to the value for **FeET6**. As for the MLCT bands, unlike the *λ*_UV_ values, the *λ*_MLCT_ values of Fe-MSPs showed a decrease with the increasing linker size, with an exceptionally low value for **FeEPh**. However, the *ε*_MLCT_ values of Fe-MSPs qualitatively showed the same trend as the *ε*_UV_ values.

As for the linker-size dependences of the *λ*_UV_, the data for the solid unimers and MSPs significantly differed from those for their solutions (Figure 2). As can be seen in Table 1, the *λ*_MLCT_ values of the solid samples are red-shifted by up to 39 nm (unimer **ET**) compared to the values for the solution. This red-shift is a measure of the extent of conformational changes (planarization increasing the overlaps of π-orbitals) of solvated molecules accompanying their packing into solid-state lattices [53], which is highly variable. For example, these red shifts observed for Zn-MSPs varied from 2 nm (**ZnTPT**), via 15 and 10 nm for **ZnE** and **ZnEPh**, to 20 nm for **ZnET** and 27 nm for **ZnET6** (see Table 1). A detailed analysis of these shifts is beyond the scope of this paper. The red shifts found for Fe-MSPs range from 3 nm (**FeE**) to 32 nm (**FeET6**). Note that the planes of near-neighboring *tpy*–unimer units in an MSP chain are ideally perpendicular to each other due to the octahedral coordination of the *tpy* groups to metal ions. The whole UV/vis spectra of Fe-MSP films are shown in Figure 3, where the wavelengths of the laser beams used to excite the Raman spectra, *λ*_exc_, are also indicated. In summary, the considerable variability of the linker effects on the spectral characteristics is indicative of the complexity of their origin.

Luminescence spectra of free unimers and their Zn-MSPs are shown in Appendix A in ESI, and the spectral characteristics determined are summarized in Table 1. The emission bands are very broad with broad, flat maxima, indicating the simultaneous presence of many different chromophores in the emitting layers. The emission band of **ZnET** films also showed a normalized intensity of 0.8 up to about 750 nm and an intense tail extending deep into the NIR region. Notably, the solution spectra of Zn-MSPs range deeply into the NIR region. The Stokes shifts of unimers and Zn-MSPs range from ca. 4000 to 5000 cm^−1^ (Table 1). They are roughly about 1000 to 2000 cm^−1^ larger than the shifts observed for the Zn-MSPs derived from unimers with oligophenylene main-chains and EPh linkers to tpy end-groups [26].

### 3.2. Time Changes in Raman Spectra of Films of Fe-MSPs with Phosphole Units

Fe-MSPs derived from *tpy*-type unimers generally show such weak luminescence that their Raman spectra (RS) are well-measurable. The strong luminescence of Zn-MSPs makes measuring their RS impossible. The resonance RS (RRS) were excited at wavelengths *λ*_exc_ of 445, 532, and 633 nm (see Figure 3b), currently using the laser power at the sample of 0.1 mW. The off-resonance RS (*λ*_exc_ = 780 nm) were collected with 50 times stronger excitation (5 mW). One of us noticed that with a longer acquisition time of the RRS of Fe-MSP, the spectral pattern visibly changes during the acquisition. To verify this, a set of consecutive RS was measured for each Fe-MSP sample, each with an acquisition time of 15 s, until spectral changes were evident (5–10 min). Examples of these sets obtained for with different *λ*_exc_ are shown in Figure 4 (for other examples, see Appendix A in ESI). The spectra were primarily analyzed by the subtraction method. The spectra collected in the first and last 15 s of exposure are labeled ***F*** and ***L***, respectively. The difference spectra (DS) obtained by subtracting ***L*** from ***F*** identify changes in the spectral pattern caused by the total radiation load of the sample. As can be seen, the off-resonance RS (780 nm) gave a structureless DS, indicating only the attenuation of the luminescence background. In contrast, the resonance RS gave DS with well-resolved bands, indicating that the radiation induced some structural changes in Fe-MSP films.

The richest and strongest DS provided the RRS taken with *λ*_exc_ = 532 nm, though this line is much further from *λ*_MLCT_ of **FeET** (615 nm) than *λ*_exc_ = 633 nm. However, the MLCT bands of conjugated Fe-MSPs are significantly contributed to, with transitions in the central parts of unimer units, shifting *λ*_MLCT_ to longer wavelengths [19,28]. However, *λ*_exc_ of 532 nm is near to the *λ*_MLCT_ = 552 nm of [Fe(H-*tpy*)_2_]^2+^ (H-*tpy* stands for unsubstituted terpyridine) [46,50]. This indicates that the first step of the revealed photo-induced phenomenon is the excitation of [Fe(*tpy*)_2_]^2+^ links. The DS obtained with *λ*_exc_ = 532 nm are analyzed in detail in the following sections.

### 3.3. Analyses of the Spectral Changes

**Identification of stable and unstable spectral components.** The simplest way to obtain the spectral components that changed or did not change during some process is by the method of weighed subtraction of representative spectra, which mostly is a part of the spectrometer operating software. In this procedure, the first difference spectrum (***D)*** is obtained by subtracting the last measured spectrum (***L***) from the first measured spectrum (***F***) of the given series:***D*** = ***F*** − *a**L***(1)

Here, *a* is the subtraction coefficient obtained iteratively by observing the changes caused by the subtraction, such that no negative peaks occur in the ***D*** spectrum. Since ***F*** includes all initial bands, while ***L*** includes only the bands that survived, ***D*** represents the spectral component disappearing under the excitation radiation. Accordingly, the second difference spectrum (***S****)* is calculated using the formula:***S*** = ***F*** − *b**D***(2)
represents the spectral component that remained intact by the excitation radiation. (The subtraction coefficient *b* is obtained similarly as coefficient *a*). The source spectra ***F*** and ***L***, together with the obtained spectral components, are shown in the upper section of Figure 5.

The whole spectral sets were more accurately processed by the ‘factor analysis’ method. All spectra in the set were first baseline-corrected by the method of orthogonal differences (see Appendix A for the reliability of this correction), and the resulting set was subjected to the factor analysis using the ‘singular value decomposition’ algorithm [54]. This method generally provides two or more subspectra and a set of coefficients from which any spectrum of the set can be reconstructed (for details see ESI). The analyses showed that any spectrum of a given set can be reconstructed with sufficient accuracy from just two subspectra. Linear combinations of the two subspectra, which showed no negative band, gave the subspectra of pure spectral components (***D*_FA_** and ***S*_FA_**) that agreed well with the spectra (***D*** and ***S***) obtained by the subtraction method. The structures lying behind the pure spectral components were estimated by comparing the spectral components with the spectra of related compounds, which are shown at the bottom of Figure 5. The disappearing component (***D***; ***D*_FA_**) closely resembles the RS of 2,5-di(2-thienyl)-1-thioxophospholes [41,43,44] and of the diiodo precursor of new unimers. This spectral component can, therefore, be assigned to the **TPT**-type central parts of unimeric units. The stable spectral component (***S***; ***S*_FA_**) agrees well with the RRS of [Fe(T-*tpy*)_2_]SO_4_ and other [Fe(R-*tpy*)_2_] complexes, so it can be assigned to the *tpy* units in octahedral bis(*tpy*)Fe linkages. 

**Kinetics and reversibility of the changes.** The baseline-corrected spectra of the whole set were deconvolved to the contributions of individual bands (Figure 6), and the time dependences of the integral band intensities were plotted (Figure 7). The plots clearly divided the bands into stable ones, marked with *, which evidently belong to the coordinated *tpy* units, and the disappearing ones, which showed a narrow range of mean lifetimes (44 ± 3 s), confirming that they belong to the same constitutional unit: the central **TPT**-type unit. (See also Appendix A with similar data for **FeEPh**).

The reversibility of spectral changes was examined by measuring RS from the same sample site, after a time delay during which the sample was not irradiated. The stability of the sample position in the spectrometer was checked with an optical microscope before and after each pause. Representative results of the recovery experiments are shown in Figure 8. Spectrum recovery was observed for all MSPs with a thioxophosphole unit. Notably, the RS of **FeTPT** were recovering significantly faster than the spectra of new Fe-MSPs with linkers to *tpy* groups.

The effect of the excitation beam intensity on the reversibility of spectral changes was also tested. The reversibility of spectral changes was found to decrease until it disappears with increasing intensity of the excitation beam. The Fe-MSPs behaved differently than the Fe^2+^ complexes of monotopic *tpy* ligands. The images of films of various samples treated with an intense laser beam (5 mW) are shown in Figure 9. The **FeET** and **FeTPT** films showed stable greenish-yellow traces of illuminated sites, while the [Fe(H-*tpy*)_2_]SO_4_ complex was partially ablated and partially carbonized.

## 4. Discussion

### 4.1. Structures Responsible for the Spectral Changes

The obvious participation of [Fe(*tpy*)_2_]^2+^ groups raised the question of whether or not the found spectral changes originate from them. We did not notice such spectral changes in earlier studies of conjugated Fe-MSPs derived from α,ω-bis(*tpy*) unimers [18,25,28,29,30,55,56,57]. However, to be sure, we checked the behavior of related simple Fe complexes with monotopic *tpy* ligands and **FeTt** MSP from our recent study [15] (Figure 10). No change in spectral pattern, only a decrease in the spectral intensity due to ablation, was recorded (especially for complexes with monotopic *tpy* ligands). Thus, it is clear that Fe(*tpy*)_2_ groups alone do not cause the revealed spectral changes and that the presence of 1-thioxophosphole or a related unit in the active structure is the second necessary condition. (See also Appendix A).

### 4.2. Mechanistic Considerations

*Mere warming* of the measured MSP can be ruled out as the cause of the revealed changes, because, in general, it only non-specifically increases the spectral background and broadens Raman bands [58,59]. *The phosphole-to-phospholene ring isomerization* consisting in the shift of one double bond from the phosphole ring into the exo position yielding a cyclohexene ring has also been considered. However, this isomerization is known to be irreversible [35,37,38], which contradicts the reversibility observed. This isomerization can be, thus, also ruled out as the cause of the observed phenomenon. The *spin-crossover process* (low-spin to high-spin state transition) of Fe ions that consists in the transition of two electrons from the bonding to antibonding orbitals was also excluded, though it might occur even at room temperature [60,61,62]. Such transition distorts the central octahedron by mainly extending the metal-to-ligand bonds with vibrations that lie at low frequencies (due to the high mass of Fe ions) [63]. However, all significant spectral changes occurred at much higher frequencies and relate exclusively to the bands of 1-thioxophosphole units, not *tpy* units. Moreover, this process in iron complexes is known to have specific steric requirements and ligands such as those by pyrazole or tetrazole rings [64]. Thus, the spin-crossover process can be also ruled out as the cause being sought.

*Photoinduced oxidation of Fe(II) ions* seems to be the most probable cause of the observed changes. The mere existence of the long-time stable MLCT band associated with the [Fe(*tpy*)_2_]^2+^ units proves a reversible partial transfer of electrons from Fe^2+^ ions into *tpy* ligands, i.e., reversible partial oxidation of these ions even in daylight. Therefore, excitation into this band can be expected to enhance this transfer. If the transferred electrons are not consumed in a ligand modification, the overall photo-enhanced MLCT should remain reversible. Simple [Fe(*tpy*)_2_]^2+^ complexes and Fe-MSPs without thioxophosphole rings behaved this way, but Fe-MSPs with these rings did not. The strongly electron-withdrawing group P = S can be considered the reactive center responsible for this exceptional behavior. This group has already been proposed as the cause of the intramolecular charge transfer from a thiophene ring into the adjacent thioxophosphole ring [43]. Moreover, DFT calculations [19,43,44] showed that the electron density of the LUMO of the TPT structures is mainly localized on the thioxophosphole ring. Thus, the most likely scenario is the reduction of the P atom by the transferred electron to form a sulfide anion, which compensates for the increased charge of the iron ion (Figure 3).

However, it should still be explained why this transformation is considerably reversible at low exposure but irreversible at intense exposure. The following factors have to be considered: (a)The motion of an electron in a solid lattice is associated with the lattice deformation, mostly acting against this motion [53,65].(b)Repeating units with newly formed ions Fe^3+^ and S^−^ represent “structure islands”, with the distribution of localized charges significantly differing from the original distribution in the surrounding intact domains, which introduces instability into the lattice. The greater the number of new ions in the original lattice, the less stable this lattice is.(c)Stabilization of the disrupted lattice requires either restoring its original structure or creating a new structure by conformational changes and displacements of mobile counterions, which is not easy in a solid.

The above points can be summarized in the mechanism shown in Figure 4.

At low-intensity excitation, the frequency of electron transfers from Fe^2+^ ions to unimeric units will be low, and, thus, the lattice disturbance will also be small. In such a case, the dominant original structure will successfully counteract the transformations that are unfavorable to it, and the electron transfer process should be essentially reversible. The increase in excitation intensity will increase the population of new ions in the lattice, until the disruption of its original structure becomes so large that the transition to a new structure optimal for the presence of new ions stabilizes the lattice and the photo-enhanced MLCT process becomes irreversible.

## 5. Conclusions

While the replacement of the central thiophene ring with a phosphole ring in a molecule of the α,ω-bis(*tpy*)terthiophene unimer caused a substantial red-shift of the unimer optical band (∆*λ*_UV_ = 75 nm) [19] but no increase in the molar absorption coefficient *ε*, the incorporation of conjugated linkers between the TPT-type central unit and *tpy* end-groups caused only a small red shift of *λ*_UV_ (+15 to 25 nm) but a substantial increase in the unimer *ε* value up to a more than twofold value, depending on the linker structure. Moreover, the linkers also significantly extended the unimer luminescence emission into the NIR region, but they did not improve the luminescence quantum efficiency. The closeness of the *λ*_UV_ values and the difference in the *ε* values also remained reduced, to an extent, in Zn- and Fe-MSPs. Quite remarkable is the fact that the highest *λ*_MLCT_ value exhibited the Fe-MSP without linkers, **FeTPT**, which nevertheless also showed the lowest *ε* value.

The most remarkable part of this article is undoubtedly the discovery of the reversible disappearance of the bands of the central phosphole unit in the resonance Raman spectra of Fe-MSPs, since a similar phenomenon has not been described so far. The fact that the phenomenon was exhibited exclusively by Fe-MSPs in which the MLCT process occurs and that the phenomenon is induced by radiation with a wavelength falling within the absorption MLCT band logically leads to the hypothesis that the phenomenon is related to the MLCT process. We experimentally demonstrated that the mere presence of the MLCT is a necessary but not a sufficient condition for the observed phenomenon to occur. We also found that none of the FeMSPs exhibit this process in a solution. This suggests that inhibited conformational changes can play a crucial role in the process. These experimental facts became the basis of the proposed hypothesis about the mechanism of the discovered phenomenon. We are, naturally, aware that this is a primary hypothesis, the possible proof of which will require additional experimental data and theoretical calculations, though we do not have the needed expertise for implementing them. Relatively large electronically excited systems with iron ions are not at all easy to perform calculations on.

## Data Availability

Not applicable.

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
