# Peer review of "Novel Metallo-Supramolecular Polymers with 1-Thioxophosphole Main-Chain Units and Remarkable Photoinduced Changes in Their Resonance Raman Spectra"

_polymers, 2022, doi:10.3390/polym14235207_

Round 1

Reviewer 1 Report

Vohlidal and coworkers reported a series of metallo-supramolecular polymers, and the damping of their photo-induced Raman spectra is highly reversible at the low excitation laser power, but gradually becomes irreversible as the power reaches. However, such change is only characterized by spectroscopy is not enough, and more systemic investigation should be provided. I recommend the major modification before acceptance of this paper.

1.    For synthesis section, the HRMS images should be provided.

2.    These polymers structure characterization is needed, like FTIR, XPS, TGA, etc.

3.    After addition of metal ion, the MLCT is observed. Thus, the coordination constant between metal ion and tpy unit is needed to confirm the molar ratio of 1:2.

4.    More tools (like XPS) are helpful for characterization of structural changes under excitation of laser power.

5.    Please explain the influence of conjugation of metallo-supramolecular polymers on the excitation wavelength of laser power for photo-induced changes in their resonance Raman spectra.

6.    The photo-induced reversible phenomenon is interesting, and its potential application should be carried out in this work.

7.    Many other works about terpyridine-metal-based coordination polymers should be cited, like Nat. Commun. 2015, 6, 6713; J. Am. Chem. Soc. 2017, 139, 5359–5366; J. Mater. Chem. C 2019, 7, 9159–9166; ACS Appl. Mater. Interfaces 2020, 12, 35181–35192; Polymers 2021, 13, 1002.

Reviewer 2 Report

The manuscript needs minor revision before publication. So please consider the following comments:

1. The abstract part must be modified. Just put the main goals and results.

2.The quality of all figures must be improved.

3. The conclusion part must be inserted at the end of text.

4. The text must be checked in case of typos, grammatical errors and spaces.

Round 2

Reviewer 1 Report

no more comments